# ROM1 is redundant to PRPH2 as a molecular building block of photoreceptor disc rims

**Tylor R Lewis[1]\*, Mustafa S Makia[2], Carson M Castillo[1], Ying Hao[1], Muayyad R Al-Ubaidi[2,3], Nikolai P Skiba[1], Shannon M Conley[4], Vadim Y Arshavsky[1,5]\*, Muna I Naash[2,3]\***

[1]Department of Ophthalmology, Duke University Medical Center, Durham, United States; [2]Department of Biomedical Engineering, University of Houston, Houston, United States; [3]College of Optometry, University of Houston, Houston, United States; [4]Department of Cell Biology, University of Oklahoma Health Sciences Center, Oklahoma City, United States; [5]Department of Pharmacology and Cancer Biology, Duke University Medical Center, Durham, United States

**\*For correspondence:**
tylor.lewis@duke.edu (TRL);
vadim.arshavsky@duke.edu
(VYA);
mnaash@central.uh.edu (MIN)

**Competing interest:** The authors declare that no competing interests exist.

**Abstract** Visual signal transduction takes place within a stack of flattened membranous 'discs' enclosed within the light-sensitive photoreceptor outer segment. The highly curved rims of these discs, formed in the process of disc enclosure, are fortified by large hetero-oligomeric complexes of two homologous tetraspanin proteins, PRPH2 (a.k.a. peripherin-2 or rds) and ROM1. While mutations in PRPH2 affect the formation of disc rims, the role of ROM1 remains poorly understood. In this study, we found that the knockout of ROM1 causes a compensatory increase in the disc content of PRPH2. Despite this increase, discs of ROM1 knockout mice displayed a delay in disc enclosure associated with a large diameter and lack of incisures in mature discs. Strikingly, further increasing the level of PRPH2 rescued these morphological defects. We next showed that disc rims are still formed in a knockin mouse in which the tetraspanin body of PRPH2 was replaced with that of ROM1. Together, these results demonstrate that, despite its contribution to the formation of disc rims, ROM1 can be replaced by an excess of PRPH2 for timely enclosure of newly forming discs and establishing normal outer segment structure.

## eLife assessment

This **valuable** study is focused on the requirement of the photoreceptor-specific tetraspanins, ROM1 and PRPH2, for the formation of light-sensitive membrane discs. The evidence supporting the claim that deficiency in one of the proteins can be compensated by the other is **convincing**, with both established and advanced techniques yielding results that will be of interest to those studying photoreceptor development and membrane curvature.

## Introduction

The light-sensitive outer segment organelle of vertebrate photoreceptor cells is a modified cilium containing a stack of disc-shaped membranes, or 'discs'. While initially formed as serial evaginations of the photoreceptor plasma membrane, mature discs are separated from this membrane and are fully enclosed inside the outer segment (*Steinberg et al., 1980*; *Burgoyne et al., 2015*; *Ding et al., 2015*; *Volland et al., 2015*) (reviewed in *Spencer et al., 2020*). Membrane enclosure involves the formation of highly curved, hairpin-shaped disc rims. Two homologous tetraspanin proteins, PRPH2

(also known as peripherin-2 or rds) and ROM1, localize specifically to disc rims (*Connell and Molday, 1990*; *Travis et al., 1991*; *Bascom et al., 1992*; reviewed in *Stuck et al., 2016*). Within the disc rim, PRPH2 and ROM1 form homo- and heteromeric complexes (*Bascom et al., 1992*; *Goldberg et al., 1995*; *Goldberg and Molday, 1996b*; *Goldberg and Molday, 1996a*) that are assembled into larger oligomers connected by disulfide bonds (*Goldberg et al., 1998*; *Loewen and Molday, 2000*; *Chakraborty et al., 2009*; *Zulliger et al., 2018*). Three parallel interconnected chains of these oligomers are wrapped around the circumference of the mature disc and support its hairpin-shaped structure (*Pöge et al., 2021*).

Despite a high level of structural homology between PRPH2 and ROM1 (*Li et al., 2003*), the consequences of their mutations for photoreceptor health are quite different. Around 200 mutations of *PRPH2* are shown to cause a heterogeneous set of inherited retinal diseases, including retinitis pigmentosa, cone-rod dystrophy, and macular dystrophies (*Landrum et al., 2018*; *Peeters et al., 2021*). Yet, mutations in *ROM1* typically cause digenic retinitis pigmentosa in conjunction with mutations in *PRPH2* (*Kajiwara et al., 1994*; *Dryja et al., 1997*), except for a handful of reports of mutations in *ROM1* in patients without accompanying *PRPH2* mutations (*Bascom et al., 1995*; *Sakuma et al., 1995*; *Reig et al., 2000*). With regard to animal models, the *Prph2* knockout mouse, commonly known as the *rds* mouse, has a complete failure of outer segment formation (*Cohen, 1983*; *Jansen and Sanyal, 1984*), while the *Rom1* knockout mouse forms outer segments with relatively minor structural abnormalities (*Clarke et al., 2000*). This discrepancy could be explained, at least in part, by the importance of the cytoplasmic C-terminus of PRPH2, which has been shown to retain membranes at the photoreceptor cilium, thereby allowing their remodeling into outer segment discs (*Salinas et al., 2017*). However, most of the disease-associated PRPH2 mutations spare its C-terminus, so the complete explanation of this discrepancy awaits further investigation of the role of ROM1.

In this study, we investigated the role of ROM1 in the formation of photoreceptor disc rims and the process of disc enclosure. We first revisited the phenotype of the *Rom1* knockout (*Rom1*[−/−]) mouse and made several novel observations. Using quantitative proteomics, we showed that the knockout of ROM1 causes a compensatory increase in the relative disc content of PRPH2. Despite the total tetraspanin content being essentially the same as in WT discs, *Rom1*[−/−] discs displayed a delay in their maturation and an increase in outer segment diameter. In addition, *Rom1*[−/−] discs lacked the indentations of their rims, known as incisures, present in WT discs. These changes could be explained, at least in part, by an alteration in tetraspanin oligomerization likely arising from a rearrangement of intramolecular disulfide bridge(s) in PRPH2. Strikingly, these morphological phenotypes can be rescued by transgenic overexpression of PRPH2, suggesting that ROM1 is not absolutely necessary for the formation of normal outer segments. In another set of experiments, we investigated whether disc rims could be formed in a knockin mouse in which PRPH2 was replaced by a chimeric protein consisting of ROM1 bearing the C-terminus of PRPH2. While the knockin photoreceptors failed to form orderly disc stacks, outer segment membranes preserved the ability to form hairpin-shaped rims. Together, these results demonstrate that, despite its contribution to disc formation and intrinsic ability to support disc rim structure, ROM1 can be replaced by an excess of PRPH2 for the formation of a normal outer segment.

## Results
### Loss of ROM1 causes a compensatory increase in the relative outer segment content of PRPH2

Three proteins, rhodopsin, PRPH2 and ROM1, comprise ~98% of the total transmembrane protein material in normal discs (*Skiba et al., 2023*). Recently, it has been proposed that the overall dimensions of a disc are determined by the molar ratio between the total tetraspanin content (PRPH2 and ROM1) forming the disc rim and rhodopsin forming the disc surface (*Lewis et al., 2023*). Therefore, interpreting the phenotype of the *Rom1*[−/−] mouse requires an understanding of how the loss of ROM1 affects the outer segment content of PRPH2. This is particularly important because discs deficient in PRPH2 display a compensatory increase in the relative amount of ROM1 (*Lewis et al., 2023*), suggesting that the opposite (i.e. a compensatory increase in the amount of PRPH2 when there is a deficiency in ROM1) may also be true. The latter is consistent with the report that the combined amount of PRPH2 and a knockin chimera between the transmembrane portion of ROM1 and C-terminus of PRPH2 is increased in the absence of ROM1 (*Conley et al., 2019*).

**Table 1.** Quantification of molar ratios between PRPH2, ROM1 and rhodopsin in WT and *Rom1-/-* outer segments.

| Protein molar ratios* | WT | *Rom1-/-* |
|---|---|---|
| PRPH2:rhodopsin | 1:18.1±0.5 | 1:12.2±1.3 |
| ROM1:rhodopsin | 1:36.3±3.9 | n/a |
| PRPH2:ROM1 | 2.0:1±0.3 | n/a |
| (PRPH2 +ROM1):rhodopsin | 1:12.1±0.2 | 1:12.2±1.3 |

*Values are shown as mean ± s.d. n/a: not applicable. Two outer segment preparations from mice of each genotype were analyzed.

The online version of this article includes the following source data for table 1:

**Source data 1.** Quantification of molar ratios between tetraspanins and rhodopsin in WT and *Rom1-/-* mice – raw data.

To address this possibility, we employed a quantitative mass spectrometry approach (*Skiba et al., 2023*) recently used to determine the molar ratio among PRPH2, ROM1 and rhodopsin in other mutant photoreceptors (*Lewis et al., 2023*). In these experiments, the outer segment content of PRPH2 was determined as a molar fraction of rhodopsin. Because the packing density of rhodopsin in discs is unaffected by any changes in disc dimensions (*Liang et al., 2004*), this parameter reflects the relative content of PRPH2 in each disc, independent of any changes in outer segment length or volume. We found that the relative content of PRPH2 in *Rom1-/-* outer segments increased to an ~1:12 molar ratio to rhodopsin from an ~1:18 ratio in WT outer segments (*Table 1*; see raw data in *Table 1—source data 1*). Strikingly, this 1:12 ratio is equal to the ratio between total tetraspanin (PRPH2 +ROM1) and rhodopsin in WT outer segments. Therefore, the knockout of ROM1 leads to a compensatory increase in the relative disc content of PRPH2.

## Loss of ROM1 delays disc enclosure causing increased outer segment diameter and occasional disc overgrowth

The retinal phenotype of *Rom1-/-* mice was analyzed at postnatal day 30 (P30). Our morphometric analysis showed that photoreceptor cell degeneration is just beginning at this age (*Figure 1A and B*). Interestingly, light microscopy imaging was sufficient to identify that the outer segment layer of these mice appeared disorganized and even shortened (*Figure 1A and C*). Please note that this and the three subsequent figures also include data obtained with a mouse overexpressing PRPH2 on the *Rom1-/-* background (PRPH2 OE/*Rom1-/-*), which will be described below, to facilitate a side-by-side comparison of all three phenotypes. Using transmission electron microscopy (TEM), we observed a number of outer segment structural abnormalities (*Figure 2* and *Figure 2—figure supplement 1*), consistent with those noted in the original report (*Clarke et al., 2000*). In general, outer segments appeared to be shorter and wider, with some of them displaying overgrown disc membranes not aligned in a stack (yellow arrows, *Figure 2B*). The degree of this overgrowth varied, with the extended membranes occasionally wrapping around the entire outer segment.

Because these ultrastructural defects are reminiscent of those from several PRPH2 mutant lines in which photoreceptor disc enclosure is affected (*Lewis et al., 2021*), we assessed the status of disc enclosure in *Rom1-/-* outer segments. We contrasted retinal tissue with tannic acid and uranyl acetate instead of the traditionally used osmium tetroxide (*Ding et al., 2015*). This technique yields a darker staining of newly forming, "open" discs exposed to the extracellular space than discs fully enclosed within the outer segment. This approach revealed that even outer segments lacking gross abnormalities in *Rom1-/-* mice have an increased number of open discs at their base (*Figure 3A and B*). Whereas WT rods contained a relatively constant number of ~7 open discs (consistent with previous reports *Ding et al., 2015*; *Volland et al., 2015*; *Lewis et al., 2021*), *Rom1-/-* rods had an average of ~14 open discs. In addition, the number of open discs in *Rom1-/-* rods was much more variable than in WT rods, with up to ~30 open discs occasionally observed. Lastly, we quantified outer segment diameters of *Rom1-/-* rods and found that they were on average ~35% wider than WT rods (*Figure 3C*), which is close to the ~43% increase previously reported in *Clarke et al., 2000*.

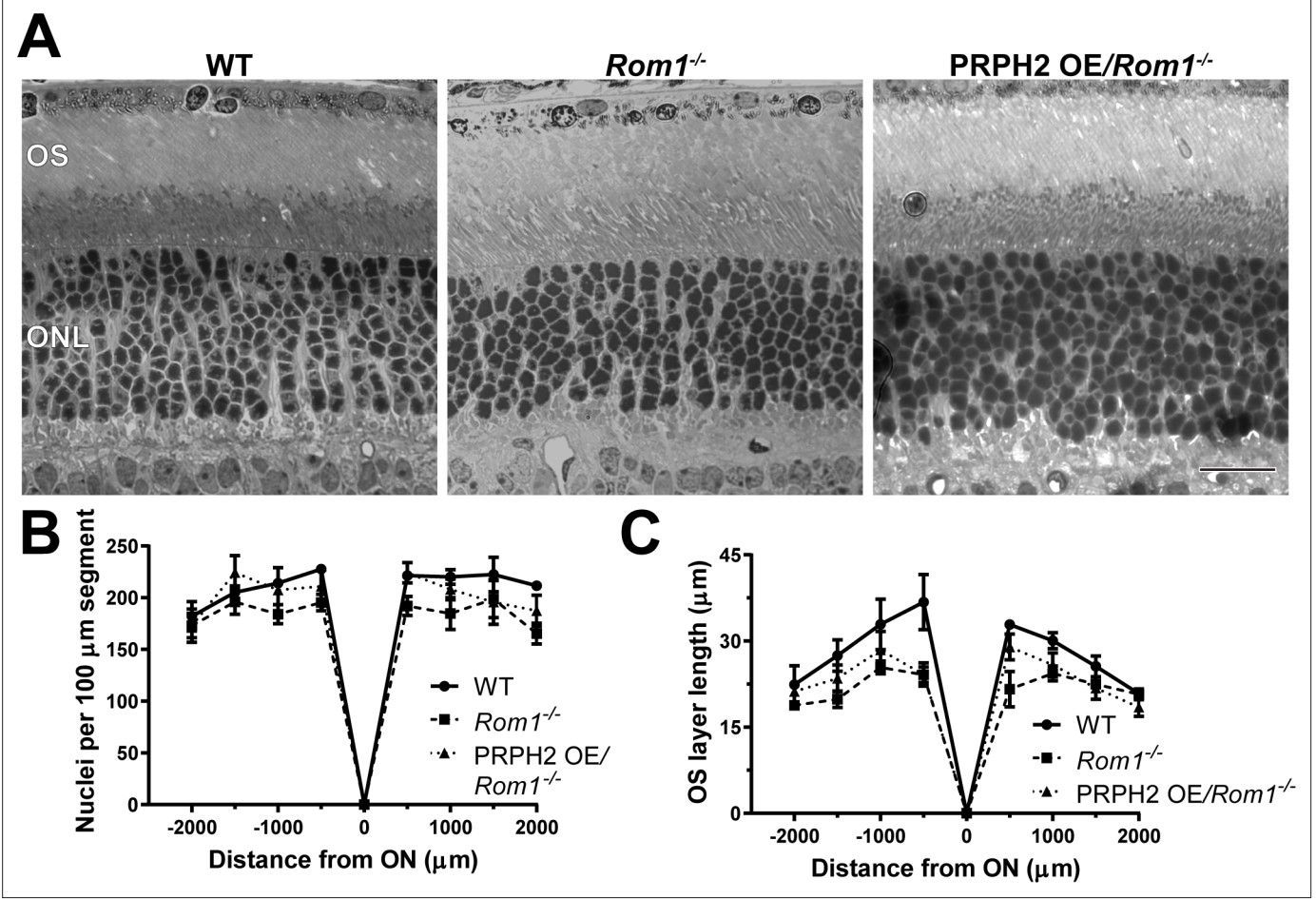

**Figure 1.** Light microscopy of retinas from WT, *Rom1⁻/⁻* and PRPH2 OE/*Rom1⁻/⁻* mice. (**A**) Representative light microscopy images of WT, *Rom1⁻/⁻* and PRPH2 OE/*Rom1⁻/⁻* retinas analyzed at P30. OS: outer segment; ONL: outer nuclear layer. Scale bar: 20 μm. (**B**) Quantification of the number of photoreceptor nuclei in a 100 μm segment of the retina at 500 μm increments away from the optic nerve (ON). Three retinas were analyzed for each genotype. Two-way ANOVA revealed statistically significant differences in the nuclear counts across genotypes (p=0.0002). Sidak's multiple comparisons post-hoc test revealed statistically significant differences between the total nuclear count between WT and *Rom1⁻/⁻* retinas (p=0.0001) and between *Rom1⁻/⁻* and PRPH2 OE/*Rom1⁻/⁻* retinas (p=0.0148), but not between WT and PRPH2 OE/*Rom1⁻/⁻* retinas (p=0.3564). (**C**) Quantification of the outer segment (OS) layer length at 500 μm increments away from the optic nerve. Three retinas were analyzed for each genotype. Two-way ANOVA revealed statistically significant differences in the OS layer lengths across genotypes (p<0.0001). Sidak's multiple comparisons post-hoc test revealed statistically significant differences for the OS layer length between WT and *Rom1⁻/⁻* retinas (p<0.0001) and between WT and PRPH2 OE/*Rom1⁻/⁻* retinas (p=0.0006), but not between *Rom1⁻/⁻* and PRPH2 OE/*Rom1⁻/⁻* retinas (p=0.2924).

These findings show that ROM1 contributes to the process of disc enclosure and that the compensatory increase in the disc content of PRPH2 does not fully replace ROM1 in this capacity. Accordingly, the increase in outer segment diameter may be explained by prolonged delivery of disc membrane material to each disc before it becomes fully enclosed. On occasion, an expansion of open discs becomes uncontrolled leading to formation of membranous whorls.

Another phenotype of *Rom1⁻/⁻* mice was revealed using TEM of tangentially sectioned retinas (**Figure 4**). Normal mouse discs contain a single deep indentation of their rims, called an incisure (yellow arrowheads, **Figure 4**). In contrast, no incisures were observed in *Rom1⁻/⁻* discs. Because the rims of both disc circumference and incisure are built from PRPH2/ROM1, the lack of incisures in *Rom1⁻/⁻* discs may be explained by the entire pool of remaining tetraspanin being deposited at the circumference of larger discs.

## ROM1 regulates the supramolecular organization of PRPH2

We next sought to address why the compensatory increase in the disc content of PRPH2 in *Rom1⁻/⁻* rods is insufficient to support normal disc morphogenesis. In this context, it is important to consider

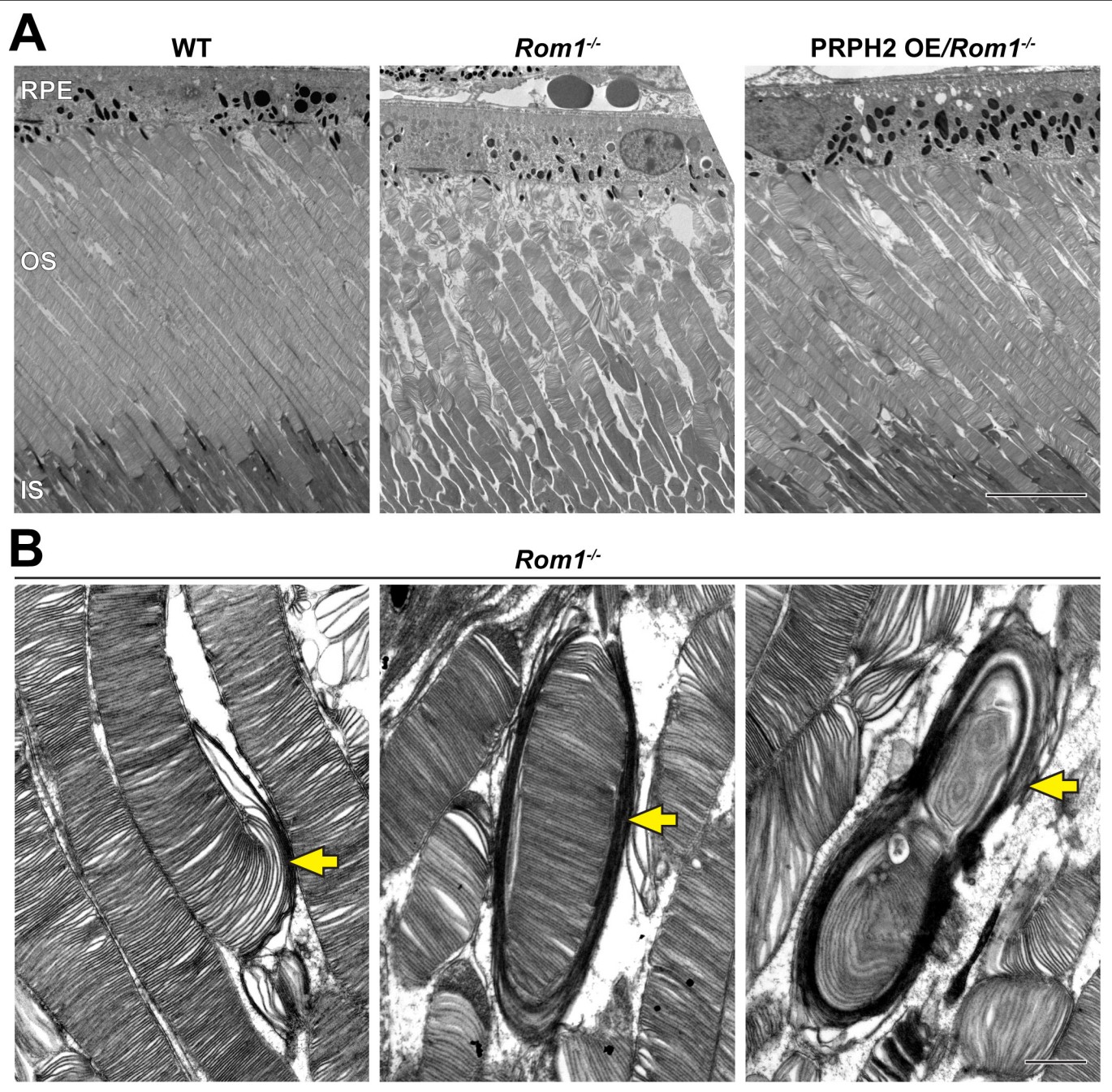

**Figure 2.** Ultrastructural analysis of retinas and rod outer segments from WT, *Rom1⁻/⁻* and PRPH2 OE/*Rom1⁻/⁻* mice. (**A**) Representative low magnification TEM images of WT, *Rom1⁻/⁻* and PRPH2 OE/*Rom1⁻/⁻* retinas analyzed at P30. RPE: retinal pigment epithelium; OS: outer segment; IS: inner segment. Scale bar: 10 µm. (**B**) Representative high-magnification TEM images of *Rom1⁻/⁻* outer segments. Yellow arrows indicate outer segment structural defects that range from slightly overgrown open discs to membranous whorls. Scale bar: 1 µm.

The online version of this article includes the following figure supplement(s) for figure 2:

**Figure supplement 1.** Ultrastructural analysis of WT and *Rom1⁻/⁻* retinas.

that defects in PRPH2 oligomerization, without an accompanying decrease in PRPH2 levels, also lead to defects in disc enclosure (*Lewis et al., 2021*). Therefore, we reasoned that loss of ROM1 may modulate the status of PRPH2 oligomerization. In fact, the idea that ROM1 may regulate the formation of high order PRPH2 oligomers has been put forward in a previous study (*Loewen and Molday, 2000*).

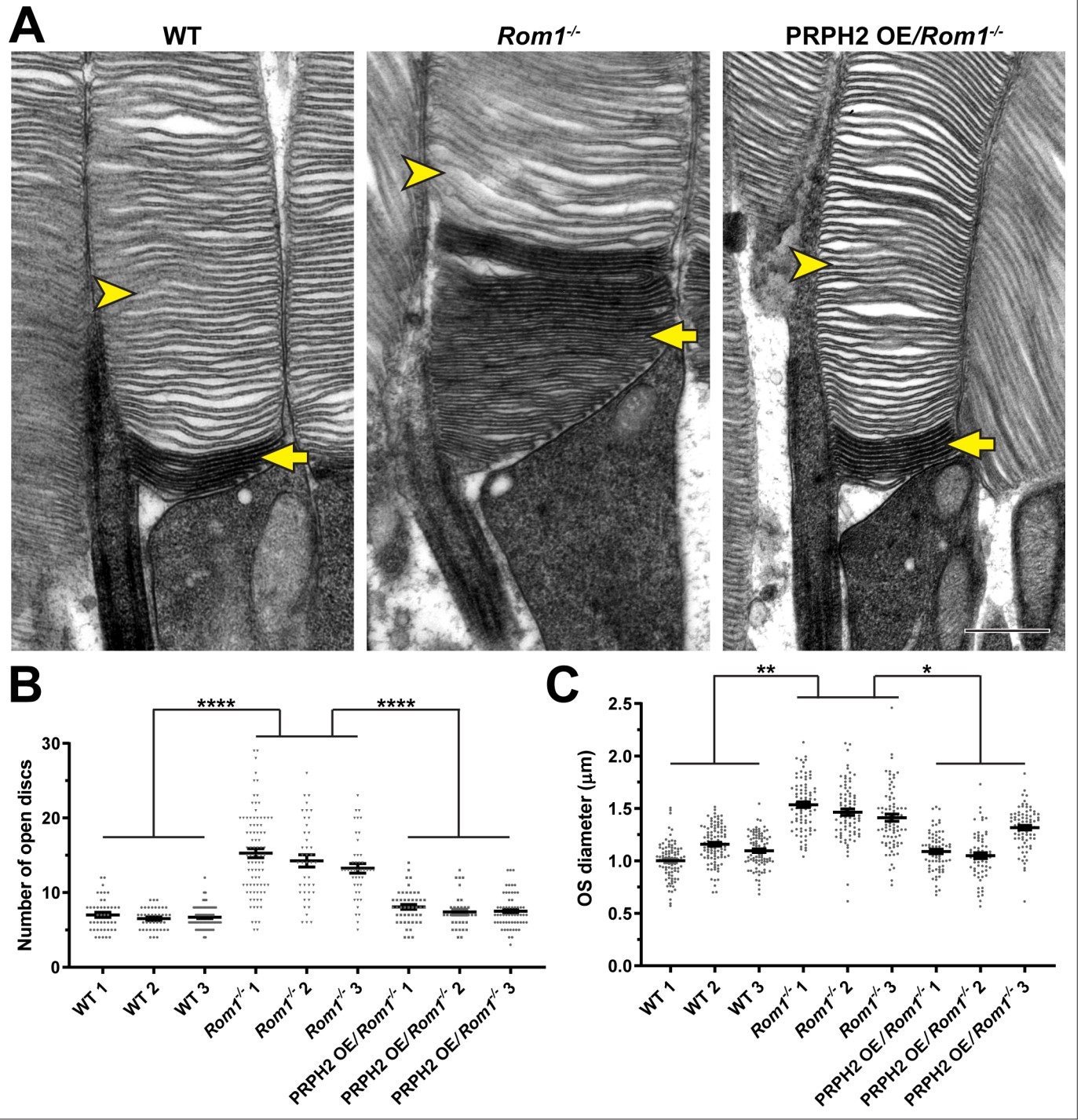

**Figure 3.** Analysis of disc enclosure in rod outer segments of WT, *Rom1⁻/⁻* and PRPH2 OE/*Rom1⁻/⁻* mice. (**A**) Representative high magnification TEM images of tannic acid/uranyl acetate-stained retinas of WT, *Rom1⁻/⁻* and PRPH2 OE/*Rom1⁻/⁻* mice analyzed at P30. This approach stains newly forming 'open' discs more intensely than mature enclosed discs. Yellow arrows point to darkly stained, unenclosed discs; yellow arrowheads point to lightly stained, enclosed discs. Scale bar: 0.5 μm. (**B**) Quantification of the number of darkly stained open discs at the rod outer segment base. Each data point represents a single outer segment. For each genotype, three retinas were analyzed with at least 35 outer segments analyzed per retina. Data were plotted with samples separated, while statistical analysis was performed on the averages within each retina (n=3 for each genotype). One-way ANOVA revealed statistically significant differences in the number of open discs across genotypes (p<0.0001). Tukey's multiple comparisons post-hoc test revealed statistically significant differences in the number of open discs between WT and *Rom1⁻/⁻* (p<0.0001) and *Rom1⁻/⁻* and PRPH2 OE/*Rom1⁻/⁻* (p<0.0001) mice, but not between WT and PRPH2 OE/*Rom1⁻/⁻* mice (p=0.2686). (**C**) Quantification of the outer segment (OS) diameter. For each genotype, three retinas were analyzed with at least 66 outer segments analyzed per retina. One-way ANOVA revealed statistically significant differences

*Figure 3 continued on next page*

*Figure 3 continued*

in the OS diameters across genotypes (p=0.0074). Tukey's multiple comparisons post-hoc test revealed statistically significant differences in the OS diameters between WT and *Rom1⁻/⁻* (p=0.0083) and *Rom1⁻/⁻* and PRPH2 OE/*Rom1⁻/⁻* (p=0.0197) mice, but not between WT and PRPH2 OE/*Rom1⁻/⁻* mice (p=0.7198). Note that these diameters were measured in longitudinal sections, in which outer segment are not always sectioned across their widest part; therefore, these values are likely under-representations of the true OS diameters. However, this does not affect the comparison across genotypes.

To address the possibility that PRPH2 oligomerization may be disrupted without ROM1, we first performed SDS-PAGE of retinal lysates from WT and *Rom1⁻/⁻* mice under non-reducing conditions (i.e. no DTT in samples). Under these conditions, PRPH2 is represented by two bands, one corresponding to ~35 kDa monomers and another to ~75 kDa dimers linked by a disulfide bond (*Figure 5A*). Both bands are somewhat diffuse, likely reflecting the heterogeneous status of PRPH2 glycosylation. The majority of PRPH2 in WT retinas is represented by dimers, consistent with its origin from large, disulfide-bound oligomers.

Two differences were observed in *Rom1⁻/⁻* retinas. First, there was an ~50% increase in the fraction of PRPH2 that runs in a monomeric state (*Figure 5A and B*), indicating that PRPH2 oligomerization may be negatively affected by loss of ROM1. Second, the monomeric form of PRPH2 was represented by a doublet, consistent with a previous observation (*Stuck et al., 2015*). This doublet was consolidated into a single band under reducing conditions in the presence of DTT (*Figure 5C*), which suggests that the two bands in *Rom1⁻/⁻* retinas represent pools of PRPH2 molecules with different patterns of internal disulfide bonds. Apart from the C150 residue involved in intermolecular disulfide bonding of a PRPH2 molecule to another PRPH2 or ROM1, there are six other conserved cysteines in PRPH2 that are involved in intramolecular disulfide bonding (*Goldberg et al., 1998*). Therefore, it is possible that ROM1 contributes to PRPH2 assuming the intermolecular disulfide bond conformation optimal for tetraspanin oligomerization at the disc rim.

To further investigate the status of PRPH2 oligomerization in *Rom1⁻/⁻* mice, we utilized a technique typically used to describe PRPH2 and ROM1 complexes of various sizes. In this approach, complexes are extracted from disc membranes in the presence of an non-ionic detergent and separated by velocity sedimentation on a sucrose gradient under non-reducing conditions (*Clarke et al., 2000*;

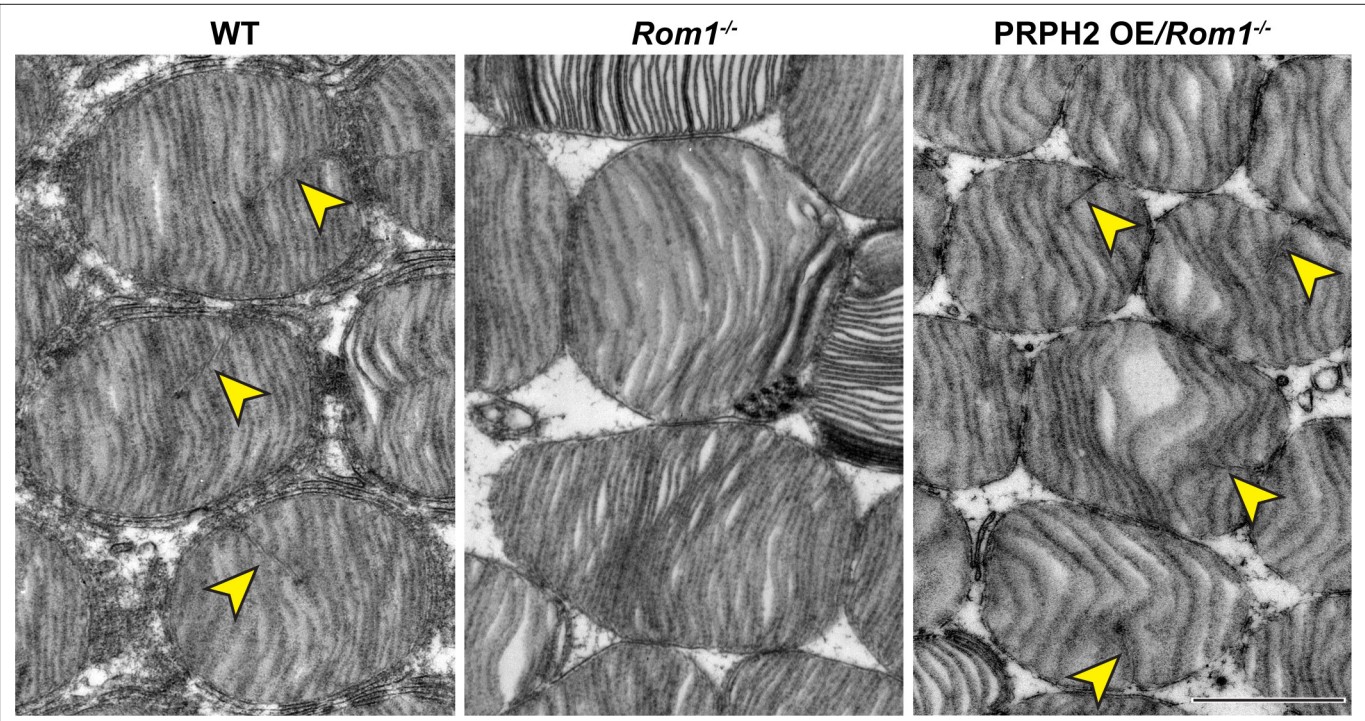

**Figure 4.** Ultrastructural analysis of disc incisures in WT, *Rom1⁻/⁻* and PRPH2 OE/*Rom1⁻/⁻* mice. Representative TEM images of retinas, tangentially sectioned through the outer segment layer, from WT, *Rom1⁻/⁻* and PRPH2 OE/*Rom1⁻/⁻* mice analyzed at P30. Yellow arrowheads indicate incisures observed in WT and PRPH2 OE/*Rom1⁻/⁻* discs, but not *Rom1⁻/⁻* discs. Scale bar: 1 µm.

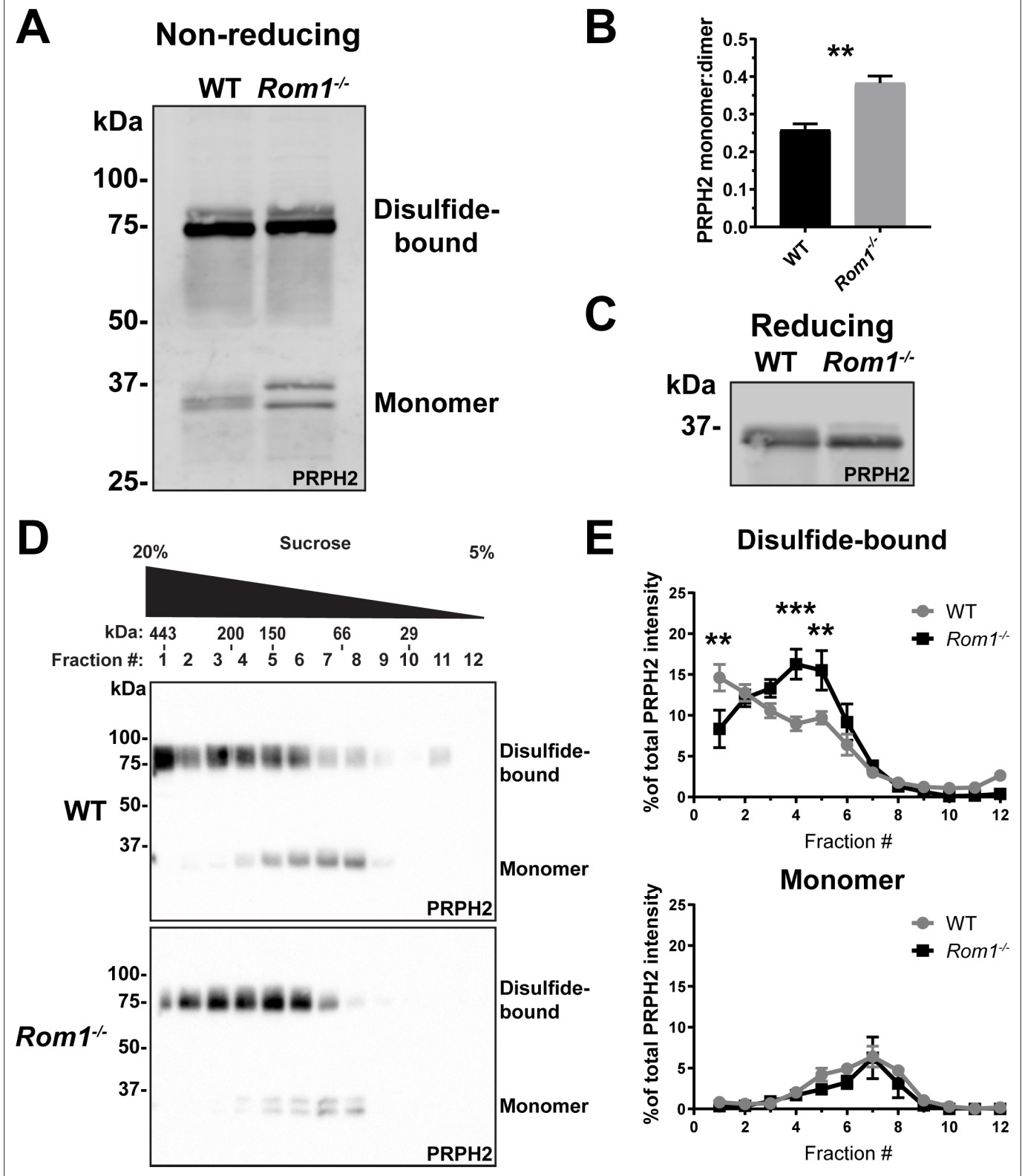

**Figure 5.** Loss of ROM1 alters the oligomerization status of PRPH2. (**A**) Western blot probed for PRPH2 after protein separation by SDS-PAGE under non-reducing conditions. Each sample contained 10 µg of lysate obtained from eyecups of WT or *Rom1⁻/⁻* mice at P30. Under these conditions, PRPH2 runs as disulfide-bound dimers (~75 kDa) and monomers (~37 kDa). The monomer band of PRPH2 runs as a doublet in *Rom1⁻/⁻* but not WT eyecups. (**B**) Quantification of the ratio between monomer and disulfide-bound (dimer) bands of PRPH2 was performed using densitometry of three independent

*Figure 5 continued on next page*

*Figure 5 continued*

lysates. For the doublet of monomer bands in *Rom1*[-/-] lysates, both bands were used for quantification. Unpaired t-test revealed a statistically significant difference in the PRPH2 monomer:dimer ratio between WT and *Rom1*[-/-] retinas (p=0.0071). (**C**) Western blot probed for PRPH2 after protein separation by SDS-PAGE under reducing conditions. Each sample contained 10 µg of lysate obtained from eyecups of WT or *Rom1*[-/-] mice. (**D**) Lysates obtained under non-reducing conditions from eyecups of WT and *Rom1*[-/-] mice were subjected to velocity sedimentation on 5–20% sucrose gradients. Twelve fractions were collected with fraction #1 corresponding to 20% sucrose and fraction #12 to 5% sucrose. Proteins from each fraction were subjected to non-reducing SDS-PAGE and Western blotting for PRPH2. The distribution of molecular mass standards across fraction, as determined in *Chakraborty et al., 2008*, is shown above the panels. (**E**) Quantification of both the monomeric and disulfide-bound bands of PRPH2 in each fraction was performed using densitometry of at least four independent samples and normalized to the total PRPH2 content across all fractions. Two-way ANOVA revealed statistically significant differences in the PRPH2 content across genotypes and fractions (p<0.0001 for disulfide-bound, p=0.9489 for monomer). Sidak's multiple comparisons post-hoc test revealed statistically significant differences between genotypes for the disulfide-bound form in fractions #1 (p=0.0022), #4 (p=0.0002), and #5 (p=0.0061).

The online version of this article includes the following source data for figure 5:

**Source data 1.** Full western blots associated with *Figure 5A*.

**Source data 2.** Full western blots associated with *Figure 5C*.

**Source data 3.** Full western blots associated with *Figure 5D*.

*Goldberg et al., 2001*; *Ding et al., 2004*; *Chakraborty et al., 2008*; *Chakraborty et al., 2009*; *Stuck et al., 2014*; *Zulliger et al., 2018*; *Milstein et al., 2020*; *Lewis et al., 2021*). We performed velocity sedimentation of lysates from both WT and *Rom1*[-/-] retinas followed by SDS-PAGE of gradient fractions under non-reducing conditions (*Figure 5D and E*). The distribution of the PRPH2 monomeric form across fractions was unaffected by the ROM1 knockout. In both WT and *Rom1*[-/-] retinas, it was most abundant in fractions #6–8, shown to correspond to PRPH2 core complexes (*Chakraborty et al., 2008*). However, the loss of ROM1 shifted the distribution of the disulfide-bound PRPH2 dimers to smaller oligomeric forms, consistent with a previous report (*Conley et al., 2019*). Taken together, these data show that loss of ROM1 does indeed affect the status of PRPH2 oligomerization and suggest that abnormal PRPH2 oligomerization underlies the defects in disc enclosure in *Rom1*[-/-] mice.

## Transgenic overexpression of PRPH2 can compensate for the loss of ROM1

We next investigated whether a further increase in the level of PRPH2 could improve the morphological defects observed in *Rom1*[-/-] outer segments. We employed a transgenic line that expresses PRPH2 in both rods and cones under control of the human IRBP promoter (*Nour et al., 2004*). While originally called NMP, we refer to this line as PRPH2 OE (PRPH2 overexpressor). In the current study, we used PRPH2 OE mice with a single copy of this transgene, which has been shown to express ~30% excess PRPH2 over WT levels (*Nour et al., 2004*). PRPH2 OE mice were crossed with *Rom1*[-/-] mice and retinas were analyzed as for WT and *Rom1*[-/-] mice (*Figures 1–3*). Strikingly, the gross morphological defects observed in *Rom1*[-/-] mice were significantly improved by transgenic overexpression of PRPH2 (*Figure 2A*). PRPH2 overexpression rescued the disc enclosure defect of *Rom1*[-/-] outer segments by restoring the number of open discs to the WT level (*Figure 3A and B*). In addition, PRPH2 OE/ *Rom1*[-/-] outer segments had normal diameters (*Figure 3A and C*) and their discs contained incisures (*Figure 4*). These data indicate that, whereas ROM1 contributes to disc formation in WT rods, it can be replaced by a sufficient excess of PRPH2.

## ROM1 is able to form disc rims in the absence of the tetraspanin body of PRPH2

While we have shown that ROM1 contributes to the formation of disc rims, it is unclear whether it can do so in the absence of PRPH2. Because disc formation is completely abolished in the absence of PRPH2, we could not address this question using PRPH2 knockout mice. Instead, we utilized a knockin mouse model, called RRCT (*Conley et al., 2019*), in which the sequence of PRPH2 is replaced with that of ROM1, except for the 64 C-terminal amino acid residues (*Figure 6A*). The resulting chimeric protein contains the C-terminus of PRPH2, required for retaining disc membranes at the outer segment base (*Salinas et al., 2017*), while the rest of the protein, required for oligomerization and functioning in disc rim formation, is ROM1.

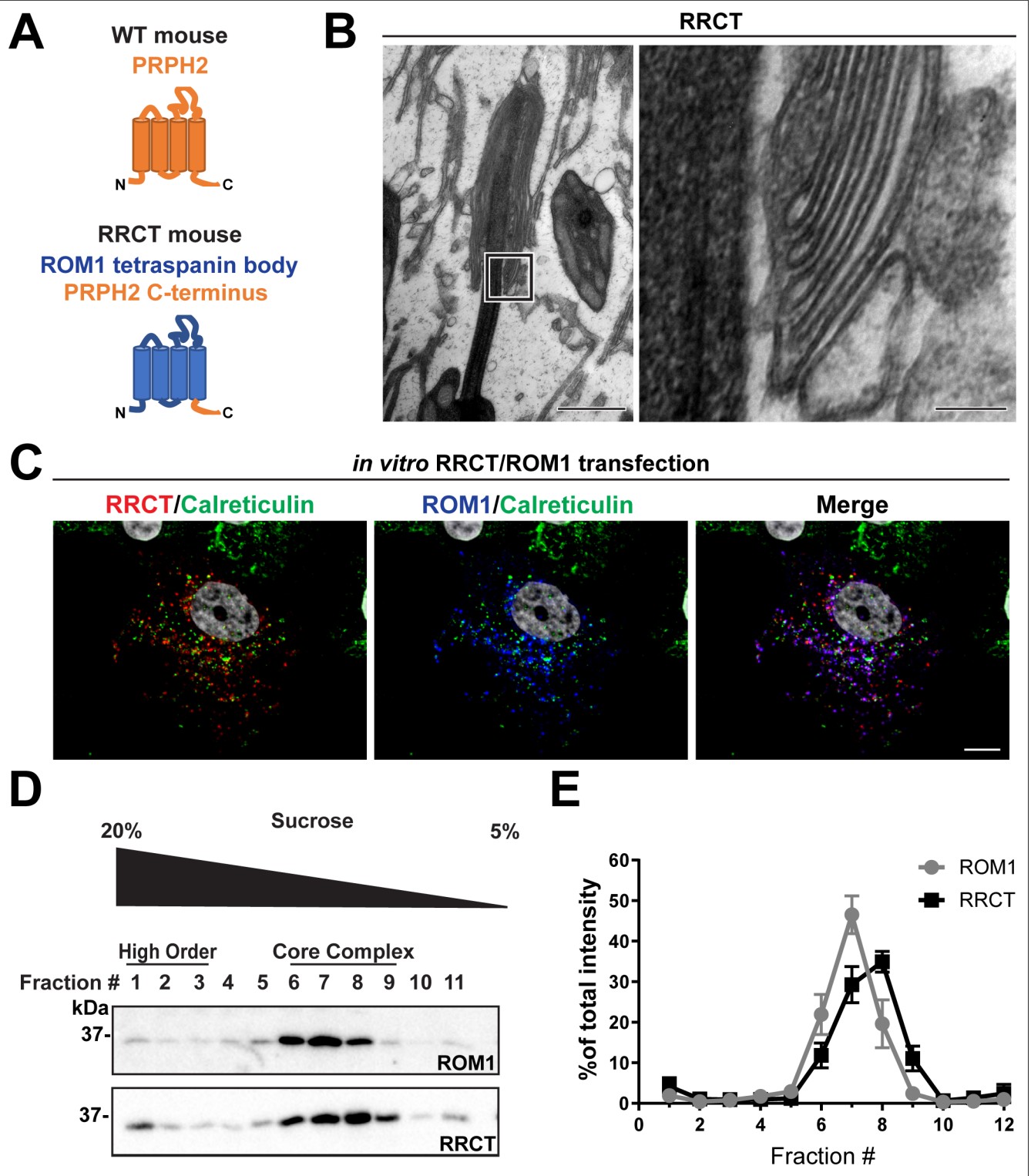

**Figure 6.** The tetraspanin body of PRPH2 can be replaced by that of ROM1 in disc rim formation. (**A**) Cartoon schematic of the RRCT chimeric tetraspanin protein. In the RRCT mouse, the *Prph2* gene has a knockin mutation that replaces it with a DNA sequence encoding a chimeric protein consisting of the tetraspanin body of ROM1 while retaining the C-terminal tail of PRPH2 that is essential for disc formation. (**B**) Representative TEM images of homozygous RRCT mice analyzed at P30. The boxed inset (left) is shown at a higher magnification (right) that reveals the presence of disc rims. Scale bars: 1 μm (left); 0.1 μm (right). (**C**) Immunofluorescent images of COS-7 cells co-transfected with RRCT-FLAG and ROM1 constructs. Cells were stained with antibodies against RRCT (red), ROM1 (blue) and calreticulin (green) to label ER membranes. Nuclei were counterstained with DAPI (grey). Scale bar: 10 μm. (**D**) Lysates obtained under non-reducing conditions from COS-7 cells co-transfected with RRCT-FLAG and ROM1 constructs

*Figure 6 continued on next page*

*Figure 6 continued*

were subjected to velocity sedimentation on 5–20% sucrose gradients. Twelve fractions were collected with fraction #1 corresponding to 20% sucrose and fraction #12 to 5% sucrose. Proteins from each fraction were subjected to reducing SDS-PAGE and Western blotting for ROM1 and RRCT. (**E**) Quantification of ROM1 and RRCT in each fraction was performed using densitometry of three independent lysates and normalized to the total content across all fractions.

The online version of this article includes the following source data for figure 6:

**Source data 1.** Full western blots associated with *Figure 6D*.

As previously reported (*Conley et al., 2019*), homozygous RRCT photoreceptors formed rudimentary outer segments (*Figure 6B*). The small number and size of these outer segment-like structures is likely explained by a very low expression of the chimeric protein (*Conley et al., 2019*), which may lead to a low efficiency of disc membrane retention. We now show that some of these structures contain stacked disc membranes that are partially enclosed within the outer segment plasma membrane. Importantly, these disc membranes have discernible rims with a characteristic hairpin-like shape. This finding indicates that ROM1 alone is, in principle, able to support the formation of disc rims in the absence of the body of PRPH2 that normally functions in this process.

In an additional set of experiments, we explored the oligomerization status of ROM1 and the RRCT chimera in the absence of PRPH2. Unfortunately, the severe disruptions of outer segments in homozygous RRCT mice precluded us from obtaining a sufficient amount of retinal lysate to perform this analysis. Instead, we used an in vitro system in which we co-transfected equal amounts of RRCT and ROM1 in COS-7 cells (*Figure 6C*). In these cells, RRCT and ROM1 co-localized in structures that were distinct from the endoplasmic reticulum, as evident from the lack of co-localization with the endoplasmic reticulum marker, calreticulin, suggesting that they were not trapped in the biosynthetic membranes. Velocity sedimentation of lysates from these cells revealed that both RRCT and ROM1 sediment in fractions #6–8 (*Figure 6D–E*), which is the same as for PRPH2/ROM1 core complexes from WT photoreceptors. These data indicate that heterologously expressed RRCT and ROM1 can form core tetraspanin complexes, even in the absence of PRPH2, that are able to support disc rim formation.

## Disc rims can be formed without disulfide bonds between tetraspanin molecules

In a final experiment, we investigated whether the formation of intermolecular disulfide bonds between tetraspanins is dispensable for disc rim formation. We have previously shown that disulfide bonds between PRPH2 molecules are not absolutely required for this function because disc rims were formed in *Prph2$^{C150S/C150S}$* knockin mice lacking the cysteine residue forming this bond (*Lewis et al., 2020*). Yet ROM1 also forms intermolecular disulfide bonds (*Chakraborty et al., 2008*), which raises the question of whether disc rim formation in *Prph2$^{C150S/C150S}$* knockin mice was driven by disulfide-linked ROM1 molecules.

To explore whether disc rims can be formed in the absence of any disulfide-linked tetraspanin molecules, we analyzed disc rim structure in *Rom1$^{-/-}$* mice bearing the *Prph2$^{C150S/C150S}$* mutation. Outer segments of these mice had morphological defects comparable to those of *Prph2$^{C150S/C150S}$* mice (*Figure 7A*). Nonetheless, orderly disc stacks were occasionally observed in these retinas. Higher magnification images of these stacks revealed the presence of disc rims which looked comparable to those in *Prph2$^{C150S/C150S}$* retinas (*Figure 7B*). This finding demonstrates that, whereas disulfide bonds connecting tetraspanin molecules are essential for maintaining the overall outer segment structure, these bonds are not absolutely required for the formation of disc rims.

In addition, staining with tannic acid/uranyl acetate revealed that the outer segments of *Prph2$^{C150S/C150S}$/Rom1$^{-/-}$* mice contained both open (yellow arrows, *Figure 7A*) and enclosed discs (yellow arrowheads, *Figure 7A*). This indicates that disulfide bonds among PRPH2 and ROM1 molecules are also dispensable for disc enclosure.

## Discussion

In this study, we analyzed the role of ROM1 in supporting the formation of outer segment discs. In addition to the previously described morphological defects of *Rom1$^{-/-}$* outer segments, we showed

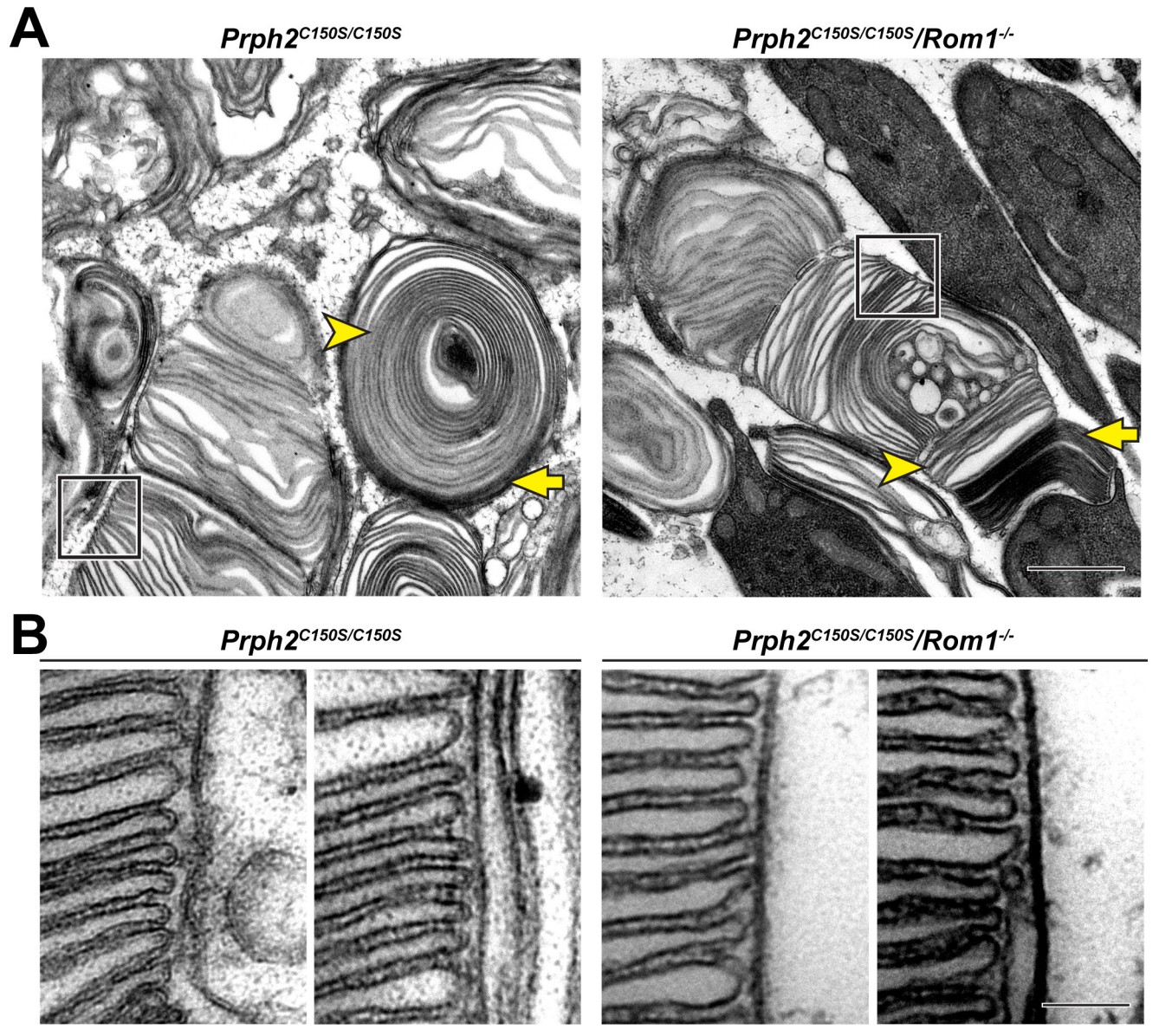

**Figure 7.** Disc rims can be formed in the absence of intermolecular disulfide bonds between PRPH2 and ROM1. (**A**) Representative TEM images of tannic acid/uranyl acetate-stained retinas of *Prph2*$^{C150S/C150S}$ and compound *Prph2*$^{C150S/C150S}$/*Rom1*$^{-/-}$ mice analyzed at P30. Photoreceptor outer segments of each mouse have severely perturbed outer segment structure. Yet, disc enclosure is not completely prevented as there are both darkly-stained, 'open' discs (yellow arrows) and lightly-stained, 'enclosed' discs (yellow arrowheads) in each genotype. The boxed region of each image depicts an area in which disc stacking appears relatively normal and permits the analysis of disc rim structure. Scale bar: 1 µm. (**B**) Higher magnification images of the regions in which disc stacking appears normal in both *Prph2*$^{C150S/C150S}$ and *Prph2*$^{C150S/C150S}$/*Rom1*$^{-/-}$ mice. Disc rims are formed in each genotype. Scale bar: 0.1 µm.

that these outer segments experience a delay in disc enclosure. This delay leads to an increase in disc diameter, loss of incisures and occasional uncontrolled disc membrane outgrowth. Remarkably, these defects can be reversed by sufficient overexpression of PRPH2, indicating a redundancy between these two tetraspanin proteins in the process of disc rim formation.

The idea of redundancy between PRPH2 and ROM1 is supported by the finding that mutations in both genetically interact to modulate disease (*Kajiwara et al., 1994*; *Dryja et al., 1997*). While individual PRPH2 mutations have greater phenotypic consequences, this could relate to both the unique role of the C-terminal tail of PRPH2 in the initial stages of disc formation (*Salinas et al., 2017*) as well

as the fact that there is twice as much PRPH2 as ROM1 in outer segments, at least in mice (*Kedzierski et al., 1999*; *Loewen and Molday, 2000*; *Skiba et al., 2023*).

While evidence that *ROM1* mutations alone may cause human visual pathology is limited (*Bascom et al., 1995*; *Sakuma et al., 1995*; *Reig et al., 2000*), photoreceptors of *Rom1*[-/-] mice eventually degenerate (*Clarke et al., 2000*). We propose that the delay in disc enclosure is the primary defect underlying this degeneration. Because discs remain open longer than normal, they may incorporate more than a normal amount of membranous material, which is continuously delivered to the outer segment. This can lead to disc membrane overgrowth and, eventually, the formation of highly dysmorphic membrane whorls. This progressive collapse of outer segment ultrastructure would eventually lead to cell death. Notably, this same progression of structural defects occurs in certain mice in which PRPH2 oligomerization is affected (*Lewis et al., 2021*).

But why is the phenotype of the heterozygous *rds* mouse containing ~50% of normal PRPH2 protein (*Hawkins et al., 1985*; *Cheng et al., 1997*) much more severe than the *Rom1*[-/-] mouse lacking 100% of ROM1? This difference is not intuitive because the molar ratio between PRPH2 and ROM1 is ~2:1 and, therefore, each mouse is expected to have ~2/3 of the normal tetraspanin content. However, there is a compensatory increase in outer segment tetraspanin in each of these two mice. We now show that relative content of total tetraspanin in *Rom1*[-/-] discs is essentially the same as in WT discs (~1:12 molar ratio to rhodopsin) due to a compensatory increase in PRPH2. On the other hand, there is only a partial compensatory increase of ROM1 in heterozygous *rds* mice (~1:15 molar ratio of PRPH2 +ROM1 to rhodopsin; *Lewis et al., 2023*) The mechanism behind these increases remains to be elucidated, but regardless, the lesser severity of the *Rom1*[-/-] phenotype could be explained, at least in part, by a larger degree of this compensatory tetraspanin increase.

Yet, the complete restoration of the total tetraspanin content in *Rom1*[-/-] outer segments does not fully rescue their morphological phenotype. This implies that, in normal photoreceptors, ROM1 contributes some unique features to the properties of tetraspanin oligomers. In fact, our data suggest that the loss of ROM1 alters both the intra- and intermolecular properties of the remaining PRPH2. We found that ROM1 likely facilitates an optimal internal disulfide bond arrangement of PRPH2 in addition to promoting the formation of larger oligomeric chains of PRPH2. Nonetheless, the outer segment deficiencies arising from loss of ROM1 were nearly completely overcome by a sufficient excess of PRPH2.

Of particular interest is our finding that discs of *Rom1*[-/-] mice lack incisures. A recent study put forth a model in which the total length of the disc rim, including the incisure, is determined by the molar ratio between the total tetraspanin protein and rhodopsin (*Lewis et al., 2023*). The lack of incisures in *Rom1*[-/-] discs may not appear intuitive since the total tetraspanin content in these discs is essentially the same as in WT discs. However, *Rom1*[-/-] discs also have an increased diameter, which allows the total rim length of WT and *Rom1*[-/-] discs to be about the same. The reason why such a tradeoff between the disc diameter and the presence/absence of an incisure exists remains to be determined, although it is likely to relate to the kinetics of disc enclosure.

In conclusion, our study shows that ROM1 makes distinct contributions to the formation of disc rims and the process of disc enclosure. Nonetheless, a sufficient excess of PRPH2 is able to compensate for the loss of ROM1, indicating that ROM1 is redundant to PRPH2 as a molecular building block of photoreceptor disc rims. Given that excess PRPH2 can rescue defects associated with loss of ROM1, it is exciting to speculate that PRPH2 overexpression can prevent the long-term photoreceptor degeneration associated with either loss of ROM1 or deficiencies arising from mutations in PRPH2 itself. It also remains to be seen whether the opposite is also true, whereby an excess ROM1 may rescue certain defects associated with mutations in PRPH2.

## Materials and methods

**Key resources table**

| Reagent type (species) or resource | Designation | Source or reference | Identifiers | Additional information |
|---|---|---|---|---|
| Strain, strain background (*Mus musculus*) | C57BL/6 J | Jackson Labs | Jax#:000664 | |

*Continued on next page*

*Continued*

| Reagent type (species) or resource | Designation | Source or reference | Identifiers | Additional information |
|---|---|---|---|---|
| Genetic reagent (*Mus musculus*) | Prph2$^{C150S}$ | *Stuck et al., 2014* | MGI:6367798 | |
| Genetic reagent (*Mus musculus*) | RRCT | *Conley et al., 2019* | | |
| Genetic reagent (*Mus musculus*) | PRPH2 OE | *Nour et al., 2004* | | |
| Genetic reagent (*Mus musculus*) | Rom1$^{-/-}$ | *Clarke et al., 2000* | MGI:2181662 | |
| Cell line (*Cercopithecus aethiops*) | COS-7 | ATCC | CRL-1651 | ATCC provides authentication and confirmation that mycoplasma contamination was not detected |
| Recombinant DNA reagent | RRCT-FLAG | *Conley et al., 2019* | | pcDNA3.1 plasmid |
| Recombinant DNA reagent | ROM1 | *Conley et al., 2010* | | pcDNA3.1 plasmid with murine *Rom1* |
| Antibody | Anti-PRPH2 (polyclonal rabbit) | *Kedzierski et al., 1999* | | WB: (1:1000) |
| Antibody | Anti-ROM1 (polyclonal sheep) | *Spencer et al., 2023* | | WB: (1:5000) |
| Antibody | Anti-RRCT (2B7; monoclonal mouse) | *Conley et al., 2014* | | WB: (1:500) |
| Antibody | Anti-ROM1 (2H5; monoclonal mouse) | *Conley et al., 2014* | | WB: (1:500) |
| Antibody | Anti-RRCT (2E7; monoclonal mouse) | *Zulliger et al., 2015* | | IF: (1:2) |
| Antibody | Anti-ROM1 (2Rom1; polyclonal rabbit) | *Ding et al., 2005* | | IF: (1:400) |
| Antibody | Anti-calreticulin (polyclonal chicken) | Abcam | ab2908 | IF: (1:250) |

## Mouse (*Mus musculus*) husbandry

Animal maintenance and experiments were approved by the local Institutional Animal Care and Use Committee (PROTO202000007; University of Houston, TX, USA) and guidelines as stated by the Association for Research in Vision and Ophthalmology (Rockville, MD). The generation of the Prph2$^{C150S}$ mouse was previously described in *Stuck et al., 2014*. The generation of the RRCT mouse was previously described in *Conley et al., 2019*. The generation of the PRPH2 OE mouse was previously described in *Nour et al., 2004*. The Rom1$^{-/-}$ mouse, described in *Clarke et al., 2000*, was generously provided by Roderick R McInnes (McGill University). All mice were genotyped to ensure that they did not contain either the rd8 (*Mattapallil et al., 2012*) or rd1 (*Pittler et al., 1993*) mutations commonly found in inbred mouse strains. All mice were on a C57BL/6 genetic background bearing the RPE65 L450 variant and housed under a 12/12 hr diurnal light (~30 lux) cycle. All experiments were performed with mice of randomized sex and, for each experiment, at least three biological replicates were analyzed.

## Transmission electron microscopy (TEM)

For all main figures, fixation and processing of mouse eyes for TEM was performed as described previously (*Ding et al., 2015*). In the afternoon, anesthetized mice were transcardially perfused with 2% paraformaldehyde, 2% glutaraldehyde and 0.05% calcium chloride in 50 mM MOPS (pH 7.4) resulting in exsanguination. Enucleated eyes were fixed for an additional 2 hr in the same fixation solution at room temperature. Eyecups were dissected from fixed eyes, embedded in 2.5% low-melt agarose (Precisionary, Greenville, NC) and cut into 200-μm-thick slices on a Vibratome (VT1200S; Leica, Buffalo Grove, IL). Agarose sections were stained with 1% tannic acid (Electron Microscopy Sciences, Hatfield,

PA) and 1% uranyl acetate (Electron Microscopy Sciences), gradually dehydrated with ethanol and infiltrated and embedded in Spurr's resin (Electron Microscopy Sciences). Seventy nm sections were cut, placed on copper grids and counterstained with 2% uranyl acetate and 3.5% lead citrate (19314; Ted Pella, Redding, CA). The samples were imaged on a JEM-1400 electron microscope (JEOL, Peabody, MA) at 60 kV with a digital camera (BioSprint; AMT, Woburn, MA). Image analysis and processing was performed with ImageJ. For *Figure 2—figure supplement 1*, fixation, processing, and imaging of mouse eyes for TEM was performed as described previously (*Stricker et al., 2005*).

## Light microscopy of histological sections

Plastic embedded blocks generated for TEM were sectioned through the optic nerve in 500 nm sections and stained with methylene blue for light microscopy as previously described (*Lobanova et al., 2008*). Images were taken with a confocal microscope (Eclipse 90i and A1 confocal scanner; Nikon) with a 60×objective (1.4 NA Plan Apochromat VC; Nikon) using Nikon NIS-Elements software. Image analysis and processing was performed with ImageJ.

## Preparation of retinal lysates

Eyecups were dissected and immediately frozen using liquid nitrogen and stored at −80 °C prior to processing. Lysates were prepared essentially as previously described (*Stuck et al., 2014*). In short, individual eyecups were lysed in 200 µl solubilization buffer (PBS, pH 7.0, containing 1% Triton X-100, 5 mM EDTA, 5 mg/ml NEM and protease inhibitors (Roche, Mannheim, Germany)). Samples were incubated on ice for 1 hr prior to being centrifuged for 30 min at 20,000 *g* at 4 °C. The supernatant was collected and subjected to either Western blot analysis directly or velocity sedimentation.

## Western blotting of retinal lysates

Protein concentration was assayed by using a colorimetric Bradford assay (Bio-Rad, Hercules, CA, USA). Western blotting was performed essentially as previously described (*Spencer et al., 2016*). In short, lysates were incubated with Laemmli sample buffer (50 mM Tris-HCl, 2% SDS, 10% glycerol and 1% Bromophenol Blue) with or without 100 mM DTT for reducing or non-reducing blots respectively. Samples containing 10 µg of total protein were incubated at 90 °C for 5 min and run on a 10–20% Tris-HCl gel. Gels were transferred onto PVDF membrane and blocked with Intercept (PBS) Blocking Buffer (Li-Cor, Lincoln, Nebraska) with 0.25% Tween-20. Blots were incubated overnight at 4 °C with 1:1000 dilution of polyclonal rabbit anti-PRPH2 C-terminal antibody (*Kedzierski et al., 1999*) and 1:5000 dilution of polyclonal sheep anti-ROM1 antibody (*Spencer et al., 2023*). After primary antibody incubation, blots were washed and incubated with 1:10,000 dilutions of donkey anti-rabbit DyLight 800 and donkey anti-sheep DyLight 680 (Invitrogen, Carlsbad, CA) for 2 hr. All experiments were repeated at least three times. Blots were imaged using Odyssey CLx imaging system (Li-Cor).

## Velocity sedimentation using sucrose gradients

Sucrose gradients were prepared as previously described (*Chakraborty et al., 2009*). In short, gradients of 5–20% sucrose were prepared by sequentially layering 0.5 ml each of 20, 15, 10, and 5% sucrose solutions in PBS with 0.1% Triton X-100 and 10 mM NEM and allowing them to sit at room temperature for 1 hr to equilibrate. Gradients were then chilled on ice for 30 min prior to loading lysate and centrifugation at 40,000 rpm on a TLS-55 swinging bucket rotor (Beckman Coulter, Brea, CA) for 16 hr at 4 °C. The bottom of each tube was pierced with a 21 G needle and 12 fractions of ~180 µl each were collected. Western blotting was performed as described above using 15 µl samples for each fraction.

## Cell culture experiments

COS-7 cells (ATCC, Manassas, VA, USA) were transfected with 7.5 µg of RRCT-FLAG and ROM1 mammalian expression constructs that were previously described (*Conley et al., 2010*; *Conley et al., 2019*) using a calcium phosphate transfection protocol as previously described (*Conley et al., 2019*). After 48 hr, cells were processed for immunofluorescent staining or velocity sedimentation as described below. Experiments were repeated three independent times.

Immunofluorescent staining was performed as described in *Conley et al., 2010*. After blocking, coverslips were incubated overnight in primary antibodies to label RRCT, ROM1 and ER membranes

as follows: (1) 1:2 dilution of monoclonal mouse anti-RRCT (2E7; described in *Zulliger et al., 2015*), (2) 1:400 dilution of polyclonal rabbit anti-ROM1 (2Rom1; described in *Ding et al., 2005*), and (3) 1:250 dilution of polyclonal chicken anti-calreticulin at 1:250 (ab2908; Abcam, Cambridge, UK). After washing, cells were incubated with 1:1000 dilution of donkey anti-mouse Alexa 568 (Invitrogen), donkey anti-rabbit Alexa 647 (Invitrogen) and goat anti-chicken Alexa 488 (Invitrogen) secondary antibodies for 1 hr. Cells were mounted in ProlongGold with DAPI (Thermo Fisher). Images were captured on a BX-62 spinning disk confocal microscope equipped with an ORCA-ER camera (Olympus, Japan) and analyzed with Slidebook 5.2 software (Intelligent Imaging Innovations, Denver, CO). Images from the spinning disk confocal microscope were deconvolved using the nearest neighbors paradigm. Images were captured with 100 x/1.40 oil objectives, and exposure times and display settings (brightness and contrast) for all images were normalized to a control section where primary antibody was omitted during processing. No gamma adjustments were made to immunofluorescent images.

For velocity sedimentation, scraped cells were lysed in the same solubilization buffer used to prepare retinal extracts. Lysates were separated as described above (200 µg protein lysate/gradient). Resulting gradient fractions were separated by SDS-PAGE under reducing conditions and blots were probed with either 1:500 dilution of monoclonal mouse anti-RRCT (2B7; described in *Conley et al., 2014*) or monoclonal mouse anti-ROM1 (2H5, described in *Conley et al., 2014*). Blots were probed with goat anti-mouse and anti-rabbit HRP secondaries (SeraCare, Milford, MA), imaged on a Bio-Rad ChemiDoc imager and analyzed in Image Lab Software version 6.0.1 (Bio-Rad).

## Quantitative mass spectrometry

A crude preparation of rod outer segments was obtained as described in *Lewis et al., 2023*. Dissected mouse retinas were vortexed in 8% OptiPrep in mouse Ringer's solution (containing 130 mM NaCl, 3.6 mM KCl, 2.4 mM $MgCl_2$, 1.2 mM $CaCl_2$, and 10 mM HEPES, pH 7.4) that was adjusted to 314 mOsm. The preparation was briefly left on ice to allow the remaining retinal tissue to settle. The supernatant was removed and centrifuged at 20,000 x *g*. Pelleted outer segments were gently washed with mouse Ringer's solution before lysis with 2% SDS in PBS. Protein concentration was measured with the Bio-Rad Protein Assay kit (Bio-Rad). Samples containing 5–10 µg of protein were mixed with 0.25–0.5 µg BSA (used as an internal standard in this analysis) and cleaved with 1 µg trypsin/ LysC mix (Promega, Madison, WI) using the SP3 beads protocol described in *Hughes et al., 2014*. The combined digest of outer segments and BSA was mixed with the digest of a chimeric protein consisting of concatenated tryptic peptides of outer segment proteins, including rhodopsin, PRPH2 and ROM1, which is described in *Skiba et al., 2023*. Mass spectrometry, data processing and data analysis were also performed as described in *Skiba et al., 2023*. For each genotype, a total of two biological replicates were analyzed.

## Experimental design and statistical analysis

For the quantification of the number of photoreceptor nuclei (*Figure 1B*), photoreceptor nuclei were counted in 100 µm boxes at 500 µm intervals from the optic nerve spanning 2000 µm in each direction for three mice of each genotype, as previously described (*Lobanova et al., 2018*). Two-way ANOVA was performed to determine statistical significance across genotype and location.

For the quantification of the number of open discs (*Figure 3B*), darkly-stained new discs at the base of the rod outer segment were counted until the first lightly-stained enclosed disc. Three3 retinas were collected of each genotype with the number of outer segments analyzed as follows: WT1, 44; WT2, 40; WT3, 68; *Rom1*[-/-]1, 87; *Rom1*[-/-]2, 42; *Rom1*[-/-]3, 44; PRPH2 OE/*Rom1*[-/-]1, 48; PRPH2 OE/*Rom1*[-/-]2, 35; PRPH2 OE/*Rom1*[-/-]3, 59. Data were plotted with samples separated, while statistical analysis was performed on the averages within each retina (n=3 for each genotype). One-way ANOVA with Tukey's multiple comparisons test was performed to determine statistical significance across genotypes. For the quantification of outer segment diameter (*Figure 3C*), three retinas were collected of each genotype with the number of outer segments analyzed as follows: WT1, 88; WT2, 97; WT3, 92; *Rom1*[-/-]1, 83; *Rom1*[-/-]2, 81; *Rom1*[-/-]3, 85; PRPH2 OE/*Rom1*[-/-]1, 75; PRPH2 OE/*Rom1*[-/-]2, 66; PRPH2 OE/*Rom1*[-/-]3, 78. Data were plotted with samples separated, while statistical analysis was performed on the averages within each retina (n=3 for each genotype). One-way ANOVA with Tukey's multiple comparisons test was performed to determine statistical significance across genotypes.

Densitometric analysis of non-saturated bands from western blots (*Figures 5 and 6*) was performed using Image Lab software v4.1 (Bio-Rad) and ImageJ. For quantification of the PRPH2 monomer:dimer ratio (*Figure 5B*), three samples were used for both WT and $Rom1^{-/-}$ mice. Unpaired t-test was performed to determine statistical significance of the PRPH2 monomer:dimer ratio between WT and $Rom1^{-/-}$ retinas. For the velocity sedimentation analysis of PRPH2 supramolecular organization (*Figure 5E*), 11 samples were used for WT and 4 samples were used for $Rom1^{-/-}$ mice. Two-way ANOVA was performed to determine statistical significance across genotypes and fractions followed by Sidak's multiple comparisons post-hoc test to determine statistical significance between genotypes in each fraction for the disulfide-bound form. For the quantification of ROM1 and RRCT in each fraction of transfected COS-7 cells (*Figure 6E*), three samples were analyzed.

All experiments were performed with mice of randomized sex. Sample sizes were determined based on previous published experiments. Data were graphed with Prism 9 (GraphPad, San Diego, CA) with error bars depicting the S.E.M. Statistical analyses were performed using Prism 9 (GraphPad). Where noted, statistical values are depicted in graphs with asterisks as follows: $p<0.05$, *; $p<0.01$, **, $p<0.001$, ***; $p<0.0001$, ****.

## Acknowledgements

This work was supported by the National Institutes of Health grants EY030451 (VYA), EY005722 (VYA), EY010609 (MIN and MRA), EY034671 (MIN), EY033763 (TRL), AG070915 (SMC) and an Unrestricted Award from Research to Prevent Blindness Inc (Duke University). The funders had no role in study design, data collection and interpretation, or the decision to submit the work for publication.

## Additional information

### Funding

| Funder | Grant reference number | Author |
| --- | --- | --- |
| National Eye Institute | EY030451 | Vadim Y Arshavsky |
| National Eye Institute | EY005722 | Vadim Y Arshavsky |
| National Eye Institute | EY010609 | Muayyad R Al-Ubaidi<br>Muna I Naash |
| National Eye Institute | EY034671 | Muna I Naash |
| National Eye Institute | EY033763 | Tylor R Lewis |
| National Institute on Aging | AG070915 | Shannon M Conley |
| Research to Prevent Blindness | Unrestricted Award | Vadim Y Arshavsky<br>Tylor R Lewis<br>Carson M Castillo<br>Ying Hao<br>Nikolai P Skiba |

The funders had no role in study design, data collection and interpretation, or the decision to submit the work for publication.

### Author contributions

Tylor R Lewis, Conceptualization, Formal analysis, Funding acquisition, Supervision, Writing – original draft; Mustafa S Makia, Carson M Castillo, Ying Hao, Supervision; Muayyad R Al-Ubaidi, Formal analysis, Funding acquisition, Supervision, Writing – review and editing; Nikolai P Skiba, Formal analysis, Supervision, Writing – review and editing; Shannon M Conley, Formal analysis, Supervision, Funding acquisition, Writing – review and editing; Vadim Y Arshavsky, Muna I Naash, Conceptualization, Formal analysis, Funding acquisition, Supervision, Writing – review and editing

### Author ORCIDs

Tylor R Lewis https://orcid.org/0000-0001-6832-7972
Vadim Y Arshavsky https://orcid.org/0000-0001-8394-3650

## Ethics

Animal maintenance and experiments were approved by the local Institutional Animal Care and Use Committee (PROTO202000007; University of Houston, TX, USA) and guidelines as stated by the Association for Research in Vision and Ophthalmology (Rockville, MD).

Reviewer #2 (Public Review): https://doi.org/10.7554/eLife.89444.3.sa1
Reviewer #3 (Public Review): https://doi.org/10.7554/eLife.89444.3.sa2
Author Response https://doi.org/10.7554/eLife.89444.3.sa3

---

# Additional files

## Supplementary files

• MDAR checklist

## Data availability

All data generated or analyzed for this study are included in the manuscript and supporting files.

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
