## [Editor Report · eLife assessment]

This **valuable** study is focused on the requirement of the photoreceptor-specific tetraspanins, ROM1 and PRPH2, for the formation of light-sensitive membrane discs. The evidence supporting the claim that deficiency in one of the proteins can be compensated by the other is **convincing**, with both established and advanced techniques yielding results that will be of interest to those studying photoreceptor development and membrane curvature.

---

## [Referee Report · Reviewer #2 (Public Review)]

In this study, Lewis et al seek to further define the role of ROM1. ROM1 is a tetraspanin protein that oligomerizes with another tetraspanin, PRPH2, to shape the rims of the membrane discs that comprise the light sensitive outer segment of vertebrate photoreceptors. ROM1 knockout mice and several PRPH2 mutant mice are reexamined. The conclusion reached is that ROM1 is redundant to PRPH2 in regulating the size of newly forming discs, although excess PRPH2 is required to compensate for the loss of ROM1.

This replicates earlier findings, while adding rigor using a mass spectrometry-based approach to quantitate the ratio of ROM1 and PRPH2 to rhodopsin (the protein packed in the body of the disc membranes) and careful analysis of tannic acid labeled newly forming discs using transmission electron microscopy.

In ROM1 knockout mice PRPH2 expression was found to be increased so that the level of PRPH2 in those mice matches the combined amount of PRPH2 and ROM1 in wildtype mice. Despite this, there are defects in disc formation that are resolved when the ROM1 knockout is crossed to a PRPH2 overexpressing line. A weakness of the study is that the molar ratios between ROM1, PRPH2 and rhodopsin were not measured in the PRPH2 overexpressing mice. This would have allowed the authors to be more precise in their conclusion that a sufficient excess of PRPH2 can compensate for defects in ROM1.

---

## [Referee Report · Reviewer #3 (Public Review)]

In this manuscript, Lewis et al. investigate the role of tetraspanins in the formation of discs- the key structure of vertebrate photoreceptors essential for light reception. Two tetraspanin proteins play a role in this process: PRPH2 and ROM1. The critical contribution of PRPH2 has been well established and loss of its function is not tolerated and result in gross anatomical pathology and degeneration in both mice and humans. However, the role of ROM1 is much less understood and has been considered somewhat redundant. This paper provides a definitive answer about the long-standing uncertainty regarding the contribution of ROM1 firmly establishing its role in outer segment morphogenesis. First, using ingenious quantitative proteomic technique the authors show PRPH2 compensatory increase in ROM1 knockout explaining the redundancy of its function. Second, they uncover that despite this compensation, ROM1 is still needed and its loss delays disc enclosure and result in the failure to form incisures. Third, the authors used a transgenic mouse model and show that deficits seen in ROM1 KO could be completely compensated by the overexpression of PRPH2. Finally, they analyzed yet another mouse model based on double manipulation with both ROM1 loss and expression of PRPH2 mutant unable to form dimerizing disulfide bonds further arguing that PRPH2-ROM1 interactions are not required for disc enclosure. To top it off the authors complement their in vivo studies by series of biochemical assays done upon reconstitution of tetraspanins in transfected cultured cell as well as fractionations of native retinas. This report is timely, addresses significant questions in cell biology of photoreceptors and pushes the field forward in a classical area of photoreceptor biology and mechanics of membrane structure as well. The manuscript is executed at the top level of technical standard, exceptionally well written and does not leave much more to desire. It also pushes standards of the field- one such domain is quantitative approach to analysis of the EM images which is notoriously open to alternative interpretations - yet this study does an exceptional job unbiasing this approach.

---

## [Author Response]

The following is the authors’ response to the original reviews.

**Reviewer #1 (Public Review):**
Summary:The precise mechanism of how tetraspanin proteins engage in the generation of discs is still an open question in the field of photoreceptor biology. This question is of significance as the lack of photoreceptor discs or defects in disc morphogenesis due to mutations in tetraspanin proteins is a known cause of vision loss in humans. The authors of this study combine TEM and mouse models to tease out the role of tetraspanin proteins, peripherin, and Rom1 in the genesis of the photoreceptor discs. They show that the absence of Rom1 leads to an increase in peripherin and changes in disc morphology. Further rise in peripherin alleviates some of the defects observed in Rom1 knockout animals leading to the conclusion that peripherin can substitute for the absence of Rom1.Strengths:A mouse model of Rom1 generated by the McInnes group in 2000 predicted a role for Rom1 in rim closure. They also showed enlarged discs in the absence of Rom1. This study confirmed this finding and showed the compensatory changes in peripherin, maintaining the total levels of tetraspanin proteins. Lack of Rom1 leads to excessive open disks demonstrated by darkly stained tannic acid-accessible areas in TEM. Interestingly, increased peripherin expression can rescue some morphological defects, including maintaining normal disc diameters and incisures. Overall, these observations lead authors to propose a model that ROM1 can be replaced by peripherin.

Thank you for your kind summary of our work.

Weaknesses:The compensatory increase in peripherin and morphological rescue in the absence of ROM1 is expected, given the previous work from authors showing (i) absence of peripherin showing increased ROM1 and (ii) "Eliminating Rom1 also increased levels of Prph2/RRCT: mean Prph2/RRCT levels in P30 Prph2+/R retinas were 34% of WT, while levels in Prph2+/R/Rom1−/− retinas were 59% of WT" from Conley, 2019. The current study provides a comprehensive quantitative analysis. However, the mechanism behind the mechanism is unclear and warrants discussion.

We referenced the result from the 2019 paper by Conley and colleagues in revision. As noted by the reviewer, new information in the current study consists of the precise quantification of the compensatory increase by a technique more accurate than semi-quantitative Western blotting. The nature of these compensatory increases is currently unknown and beyond the scope of experiments described in the current study. While this is an intriguing area for future investigation, we prefer not to speculate on the underlying mechanisms to avoid any appearance of data overinterpretation.

Photoreceptor morphology appears better when peripherin is overexpressed. Is there a rescue of rod function (assessed by ERG or equivalent measures) in peripherin OE/Rom1-/- mice? Given the extensive work in this area and the implications the authors allude to at the end, it is important to investigate this aspect.

It is indeed an interesting and potentially translationally relevant direction to address whether PRPH2 overexpression can rescue the long-term degeneration and functional defects of the loss of ROM1. Unfortunately, our work in this direction remains severely hindered by the fact that the current line of ROM1 knockout mice are notoriously poor breeders, allowing us to get only a handful of animals for each year of breeding. Therefore, we decided to limit our current study to addressing the structural roles of ROM1 and PRPH2 in supporting disc formation.

**Reviewer #1 (Recommendations For The Authors):**
Line 210: "ROM1 is able to form disc rims in the absence of PRPH2" is not demonstrated. The data shows that the tetraspanin domains are interchangeable similar to Conley, 2019. Similar concern for lines 225-226.

We agree with the point regarding the interchangeable tetraspanin domains and clarified it in the text by referring to the tetraspanin body of PRPH2 where applicable. However, the 2019 paper by Conley and colleagues did not show any ultrastructural images of disc rims in a mouse without at least one copy of WT PRPH2 being expressed. The presence of normally looking disc rims in the complete absence of the tetraspanin body of PRPH2 is an original observation of the present study.

Line 234: it is unclear what is meant by .."they are normally processed in the biosynthetic membranes" How does lack of ER localization lead to this conclusion?

We clarified this point by replacing “normally processed” with “not trapped”.

Lines 306-308: it is difficult to follow the rationale. How will a shift in the trafficking pathway affect disulfide bonds since these are formed in ER?

The reviewer makes a good point that at least the bulk of S-S bridge formation takes place during protein maturation in the ER and the ability of additional intramolecular S-S bond formation in the Golgi is questionable. We, therefore, removed this speculation from Discussion.

Given the poor development of OS, the authors could provide an estimate of how many OS-like structures were observed, with and without rims, in RRCT animals.

The gross development of outer segment structures in RRCT homozygous mice was part of the 2019 paper by Conley and colleagues. We prefer to limit repeating experiments from the previous study, but instead wanted to focus specifically on disc rim formation, which was not analyzed in RRCT homozygous mice in the previous study.

The term "function" is loosely defined throughout this manuscript. Specifically, the excess peripherin can resolve some of the morphological defects observed in Rom1 -/-, and these functional changes in morphology are the focus of this work.

We removed the word “function” in three occasions where there may be an ambiguity in its meaning, as noted by the reviewer.

Lines 115/116: Reference is missing for the statement that photoreceptor cell degeneration begins at P30.

These lines reference Figures 1A,B, which include quantification of the number of photoreceptor nuclei. These results show that ROM1 knockout retinas exhibit a modest but statistically significant degeneration at P30. The text is modified to eliminate any ambiguity.

Lines 143-144 are speculation and could be moved to the discussion section. "Prolonged delivery of disc membrane delivery to each disc" Any reference or experiments to support this statement?

We respectfully disagree with moving this short speculative sentence to Discussion. We believe that it helps the reader to follow the flow of the data, while being clearly presented as a potential explanation rather than a conclusion.

Line 245-246: Results explained in the following paragraph (247-254) do not answer the question "whether disc rim formation in PRPH2 2C150S/C150S knockin mice was driven by disulfide-linked ROM1 molecules", which is a valid and intriguing question. However, the results explained in 247-254 answer the question "if C150S PRPH2 can form discs in the absence of ROM1".

We changed the text to replace “To address this question” with “To explore whether disc rims can be formed in the absence of any disulfide-linked tetraspanin molecules”, which precisely reflects what was addressed.

**Reviewer #2 (Public Review):**
In this study, Lewis et al seek to further define the role of ROM1. ROM1 is a tetraspanin protein that oligomerizes with another tetraspanin, PRPH2, to shape the rims of the membrane discs that comprise the light-sensitive outer segment of vertebrate photoreceptors. ROM1 knockout mice and several PRPH2 mutant mice are reexamined. The conclusion reached is that ROM1 is redundant to PRPH2 in regulating the size of newly forming discs, although excess PRPH2 is required to compensate for the loss of ROM1.This replicates earlier findings while adding rigor using a mass spectrometry-based approach to quantitate the ratio of ROM1 and PRPH2 to rhodopsin (the protein packed in the body of the disc membranes) and careful analysis of tannic acid labeled newly forming discs using transmission electron microscopy.In ROM1 knockout mice PRPH2 expression was found to be increased so that the level of PRPH2 in those mice matches the combined amount of PRPH2 and ROM1 in wildtype mice. Despite this, there are defects in disc formation that are resolved when the ROM1 knockout is crossed to a PRPH2 overexpressing line. A weakness of the study is that the molar ratios between ROM1, PRPH2 and rhodopsin were not measured in the PRPH2 overexpressing mice. This would have allowed the authors to be more precise in their conclusion that a 'sufficient' excess of PRPH2 can compensate for defects in ROM1.

Thank you for these kind comments about our work. Regarding the stated weakness that we did not measure the molar ratios between PRPH2, ROM1 and rhodopsin in the ROM1 knockout line with PRPH2 overexpression: this is one experiment that we really hoped to do but were limited by the poor breeding of the ROM1 knockout line described above. With the current breeding rate, we estimate that we would need to wait for another year to get enough material to do this experiment, which we cannot do in the context of this manuscript revision. We hope, however, that eventually this may be a part of one of our future papers.

**Reviewer #2 (Recommendations For The Authors):**
The p-value for statistical significance is not listed, readers will assume the most commonly used 0.05 value was used but this should still be defined, especially since only asterisks summarizing the p-value range are provided in place of the actual p-values.

The definitions of various numbers of asterisks of significance (including p<0.05 as a minimal measure of significance) are provided in the Methods section, whereas the exact p-values are stated in figure captions.

There are 3 phrasing issues that are potentially misleading.1. While PRHP2 and ROM1 are the most abundant tetraspanins in photoreceptors they are not the only ones. It would be more precise if for example the Table 1 title was changed to 'molar ratio of outer segment tetraspanins and rhodopsin'.

We have changed the title of Table 1 to “Quantification of molar ratios between PRPH2, ROM1 and rhodopsin in WT and Rom1-/- outer segments” to be more accurate.

1. The protein expressed in RRCT mice is described as the 'tetraspanin core' while the cartoon (and original paper) shows the protein as simply being ROM1 with a different cytoplasmic C-terminus (from PRHP2). Tetraspanin core in other places is used to mean just the transmembrane bundle or that bundle with the EC loops.

We agree that the term “tetraspanin core” may be confusing. We modified the text to not use this term and, when needed, refer to this main part of the tetraspanin molecule as a “body”.

1. Line 203-205, the 'somewhat restored' qualifier should be removed. If the authors think there is an effect that is different from chance, they should use a different alpha and justify that choice.

We removed this line, as suggested.

**Reviewer #3 (Public Review):**
In this manuscript, Lewis et al. investigate the role of tetraspanins in the formation of discs - the key structure of vertebrate photoreceptors essential for light reception. Two tetraspanin proteins play a role in this process: PRPH2 and ROM1. The critical contribution of PRPH2 has been well established and loss of its function is not tolerated and results in gross anatomical pathology and degeneration in both mice and humans. However, the role of ROM1 is much less understood and has been considered somewhat redundant. This paper provides a definitive answer about the long-standing uncertainty regarding the contribution of ROM1 firmly establishing its role in outer segment morphogenesis. First, using an ingenious quantitative proteomic technique the authors show PRPH2 compensatory increase in ROM1 knockout explaining the redundancy of its function. Second, they uncover that despite this compensation, ROM1 is still needed, and its loss delays disc enclosure and results in the failure to form incisures. Third, the authors used a transgenic mouse model and show that deficits seen in ROM1 KO could be completely compensated by the overexpression of PRPH2. Finally, they analyzed yet another mouse model based on double manipulation with both ROM1 loss and expression of PRPH2 mutant unable to form dimerizing disulfide bonds further arguing that PRPH2-ROM1 interactions are not required for disc enclosure. To top it off the authors complement their in vivo studies by a series of biochemical assays done upon reconstitution of tetraspanins in transfected cultured cells as well as fractionations of native retinas. This report is timely, addresses significant questions in cell biology of photoreceptors, and pushes the field forward in a classical area of photoreceptor biology and mechanics of membrane structure as well. The manuscript is executed at the top level of technical standard, exceptionally well written, and does not leave much more to desire. It also pushes standards of the field- one such domain is the quantitative approach to analysis of the EM images which is notoriously open to alternative interpretations - yet this study does an exceptional job unbiasing this approach.According to my expertise in photoreceptor biology, there is nothing wrong with this manuscript either technically or conceptually and I have no concerns to express.

Thank you for these incredibly kind comments.

**Reviewer #3 (Recommendations For The Authors):**
I have no recommendations to make.